# Standardized protocol for laboratory rearing and breeding of the Lymnaeidae snail, *Radix natalensis* (Krauss, 1848)

Agrippa Dube[1,2]*, Chester Kalinda[1,3], Tawanda Manyangadze[1,4], Tendai Makoni[5], Moses John Chimbari[1]

1 School of Nursing and Public Health, College of Health Sciences, Howard College Campus, University of KwaZulu-Natal, Durban, South Africa, 2 Physics, Geography and Environmental Sciences Department, School of Natural Sciences, Great Zimbabwe University, Masvingo, Zimbabwe, 3 University of Global Health Equity (UGHE), Bill and Joyce Cummings Institute of Global Health, Kigali Heights, Kigali, Rwanda, 4 Geosciences Department, School Geosciences, Disaster and Sustainable Development, Faculty of Science and Engineering, Bindura University of Science and Technology, Bindura, Zimbabwe, 5 Department of Mathematics and Computer Science, Great Zimbabwe University, Masvingo, Zimbabwe

* adube168@gmail.com

## Abstract

Freshwater lymnaeid snails are involved in the transmission of fascioliasis in tropical and subtropical Africa, Asia, as well as in temperate regions. This study improved and standardized laboratory rearing and breeding of first-generation ($F_1$) *R. natalensis* using field-collected $F_0$ snails. Ninety field-collected *R. natalensis* adult snails with shell heights of 4–5 mm were divided into three experimental treatment groups: A, B, and C. Each experimental treatment group comprised of ten (10) 2L containers, with each containing 3 snails. Group A, the control, was fed *Elodea* sp. weed powder and *Cyperus papyrus* twigs from snail-sampling sites as oviposition material. Group A containers were filled with water from the snail sample sites. Spring-watered snails in experimental group B were fed with dried lettuce, fish flakes, and eggshells. In experimental group C, snails were fed with algal wafers and trout pellets in dechlorinated water. Groups B and C used polystyrene strips for oviposition. Daily snail mortality and egg mass counts were obtained. Experimental group B snails produced 69 egg masses and 500 $F_1$ offspring with the lowest snail mortality (13%). Group C produced 60 egg masses and 450 $F_1$ offspring. The mortality rate in this group was 20%. Group A control snails laid 10 eggs and 48 $F_1$ offspring. Also, mortality (66%) was higher in this group. Mean egg masses differed significantly between groups A and B (Group A: $0.85 \pm 0.22$ egg masses; Group B: $2.33 \pm 0.53$, $p = 0.034$) and A and C (Group A: $0.85 \pm 0.22$ egg masses; Group C: $2.16 \pm 0.48$, $p = 0.041$), but not between groups B and C. Treatment differences explained 11.4% ($F_{1,25} = 4.36$, $p = 0.047$) of egg mass variability. The median snail survival in group B was 8.11 days versus 4.57 days in group A. Significant differences in median survival time were observed between experimental groups (Log Rank $X^2 = 9.87$, $p = 0.007$). Group B had

**Data availability statement:** All relevant data are within the manuscript and its Supporting Information files.

**Funding:** This project has received funding from the European Union's Horizon 2020 Research and Innovation Program under grant agreement No 101000365. This work was supported the European Union's Horizon 2020 Research and Innovation Program to M J C. Agrippa Dube received monthly stipend from the European Union's Horizon 2020 Research and Innovation Program. The funders had no role in study design, data collection and analysis, decision to publish, or preparation of the manuscript.

**Competing interests:** The authors have declared that no competing interests exist.

the highest fecundity and lowest mortality among the treatment groups. However, the use of spring water increased the costs of mass breeding of snails using this approach. On the other hand, experimental group C produced a comparable number of egg masses. Thus, for mass breeding of *R. natalensis*, the use of an experimental approach from group C would be recommended as it is cheaper.

## 1. Background

Lymnaeid snails are freshwater gastropods widely distributed globally. These snails are important intermediate hosts (IHS) of trematode parasites [1]. Globally, about 20 species of lymnaeid snails have been described as potential IHS of *Fasciola* sp. [2,3]. Several lymnaeid snails serve as intermediate host snails of *Fasciola hepatica* in temperate regions of Europe, Asia, Australia, and the Americas. These include *Galba truncatula*, *Austropeplea* (*Lymnaea*) *tomentosa, Lymnaea viatrix, Lymnaea viridis, Galba cubensis,* and *Pseudosuccinea columella* [2]. In contrast, *Radix auricularia, Radix acuminata, Pseudosuccinea columella,* and *Radix natalensis* transmit *F. gigantica* in the tropical and subtropical regions of Asia and Africa [2,4]. Understanding the transmission dynamics of fascioliasis requires knowledge of its chain of transmission and the ecology of IHS involved in the transmission cycle. Understanding the ecology of these snails may also help in the improvement of control programmes [5].

Conducting laboratory mechanistic experiments to understand the ecology of snails has become an important aspect in veterinary research and public health [5]. A key requirement for conducting mechanistic experiments is raising a clean, laboratory-bred snail colony that is free from field trematode infection. When developing and rearing a lymnaeid snail colony, several factors are considered, including environmental factors such as temperature, humidity, light, and water quality, which need to be strictly monitored to ensure snail survival and reproductive success [6]. Furthermore, feed is another important factor to consider. Earlier studies have suggested lettuce and algae, supplemented with trout pellets, tropical fish flakes, and calcium for shell development, as ideal feeds for snails [7]. Most snails are hermaphrodites and reproduce through self-fertilization [8,9]. Thus, egg laying and hatching in addition to survival should be continuously monitored. Breeding of snails is also important because snails are used as hosts for research into drugs and vaccines, such as those for parasitic diseases like trematodiasis [10,11]. Laboratory-bred snails are also commonly used in controlled investigations for snail-borne disease surveillance and risk assessment [11], developing predictive disease transmission models and investigating how snails respond to various biotic and abiotic factors. Earlier malacological studies reported that *Radix natalensis*, the main intermediate host snail for *Fasciola gigantica* in the tropical and subtropical regions is difficult to culture in laboratory conditions [1]. Furthermore, breeding snails in the laboratory is critical for completing the parasite life cycles, especially when studying complicated trematodes with many hosts [1,2,12].

Little work has been done on the breeding of *R. natalensis* snails for further laboratory experimental work. To the best of our knowledge, three experimental works have been previously done to determine the influence of *F. gigantica* on the life history traits of *R. natalensis* in East and West Africa [13–15]. Earlier studies provide important insights on how to breed snails using dried lettuce, trout pellets, and crushed chicken eggshells as snail feed, as well as the use of dechlorinated water in researches conducted in Kenya and Nigeria by [7,14]. However, differences in altitude between the regions where these studies were conducted and our study area suggest that assessing the breeding feasibility of locally available snails would improve preparedness for fascioliasis outbreaks, particularly in Southern Africa. An earlier study on the breeding of *Bulinus globosus* snails suggested that the use of field-collected snails in laboratory studies may lead to inaccurate and non-reproducible results, due to the change in conditions and inadequate acclimatization time for snails [5]. *R. natalensis* is a major IHS of fascioliasis, a disease of public and veterinary health importance that affects live-stock and humans in the tropics and subtropics regions worldwide [16–19]. In livestock, infection leads to animal weight loss, reduced productivity and fertility, and reduced milk production, leading to economic losses [20]. Bovine fascioliasis is of concern in livestock production in South Africa [21,22]. Furthermore, humans get accidentally infected by consum-ing metacercariae-contaminated water crest [18]. In South Africa, human fascioliasis is not a burden, only 3 cases were reported in Gauteng (1956) and Western Cape Province (1964) [23]. To enable current and future laboratory mass breed-ing of *R. natalensis* for laboratory experiments, we reported results from a cost-effective standardized rearing and breed-ing protocol for mass-producing $F_1$ *R. natalensis* snails.

## 2. Materials and methods

### 2.1. Snail collection and identification

To enable the development of laboratory breeding $F_1$ *R. natalensis*, an $F_0$ parental generation of snails was collected from Kwafik'suthe dam (−29.671, 29.875) in Impendle municipality and The Plains farm (−29.149; 30.008) in Mpofana munic-ipality in uMgungundlovu district in KwaZulu-Natal, South Africa. Snails for the study were identified using morphological identification keys designed by Brown (1994) [24] and Appleton (1996) [25]. *R. natalensis* were identified with the shells being dextral and oblong and four rapidly increasing whorls, basal whorl usually markedly swollen. The columellar margin of the aperture twisted, and the sharp outer lip. The spire was depressed and about half the height of the aperture. The shells were colorless, yellow or dark.

### 2.2. Culturing of snails

One hundred and twenty snails were collected from the field. However, during acclimatization to laboratory conditions, 30 snails died before allocation to treatment groups. The remaining ninety snails were randomly assigned to three exper-imental groups, with the experiments being carried out at the Biomedical Resource Unit (BRU) laboratory, Westville campus of the University of KwaZulu-Natal, South Africa. Each treatment group consisted of 10 transparent 2L plastic containers (26.65 cm × 19.3 cm × 6.45 cm), with three snails per container, totaling 30 snails per group. Group A (control): fed with *Elodea* weed powder; *Cyperus papyrus* twigs provided oviposition substrate; field-collected water was used, Group B: fed with dry lettuce, fish flakes, and crushed eggshells; spring water was used, and Group C: fed with algae wafers and trout pellets; dechlorinated tap water was used (left to stand for four days). All groups were fed three times per week, and water was changed at feeding to prevent waste buildup. Snail mortalities and egg masses were recorded daily. Egg masses laid on substrates or container walls were carefully removed and transferred to new containers containing the same water type as the original group to ensure consistency. Ambient and water temperatures were monitored daily and maintained at 26 ± 1 °C (air) and 22 ± 1 °C (water), respectively. Artificial light was provided in the breeding room using a 4 × 100watt fluorescent tube. The breeding room maintained a 12-hour day and night. No artificial aeration was done in the breeding containers. Furthermore, water pH was maintained between 7.40 and 9 and checked using a Hanna multiparam-eter water testing meter.

## Ethical standards

Research permits were obtained through the Animal Research Council Committee (AREC) of the University of KwaZulu-Natal. Protocol Reference number: AREC/00005642/2023(00020552) in accordance with the South African national guidelines on animal care, handling, and use for biomedical research.

### 2.3. Statistical analysis

Data analysis was performed using StataNow/SE 18. Data on egg masses was square root transformed to enhance homogeneity of variance [5]. A one-way ANOVA was used to determine if the treatment to which snails had been exposed to had an effect on the square root transformed number of egg masses laid. Furthermore, a linear regression model on transformed egg masses was used to determine the influence of treatment on the number of egg masses laid. Kaplan Meier survival analysis was used to determine the effect of the differences that treatments had on snail survival, while the survival times were compared between the treatments using the Log Rank (Mantel-cox) tests. For all statistical tests the level of significance cut off was set at $p < 0.05$.

## 3. Results

Group A snails (fed Elodea weed powder, with papyrus twigs as oviposition substrate and field water) experienced a high mortality rate (67%), followed by Group C (20%), and Group B (13%) (Table 1). Group B snails (fed dry lettuce, fish flakes, eggshell powder, with white polystyrene strips as oviposition substrate and spring water produced the most egg masses (n=69, followed by Group C (n=59) and Group A (n=10). Group B produced the highest number of hatched juveniles (n=500), followed by Group C (n=450) and Group A (n=48) (Table 1).

### 3.1. Fecundity

Overall, 138 egg masses were collected from the experimental groups. Group A snails laid the least number of egg masses (n=10) 7.24%, followed by those in group B (n=69) 50%, and group C (n=59) 42.76%. The overall mean number of egg masses across all treatments was 5.11 (95%CI: 2.78–7.45). Fig 1 shows the mean number of egg masses laid across treatment groups.

Statistical differences in the mean number of eggs laid across the three treatments were observed ($F_{2,24}$=3.29, $p$=0.047). The mean number of egg masses laid by snails maintained in experimental group A (0.85±0.22egg masses) was lower than that of snails in experimental groups B (2.33±0.53 egg masses, $p$=0.034) and C (2.16±0.48 egg masses, $p$=0.041). On the other hand, no statistical differences in the mean number of egg masses laid were observed between experimental groups B and C ($p$=0.989) (Table 2). Furthermore, treatment type influenced the mean number of eggs laid (Coeff: 0.657, $p$=0.047), and also accounted for 11.4% of the total variability observed in the number of egg masses laid.

**Table 1. Summarized results of indicators performance of snails in different treatments.**

| Group | Group A (Control) | Group B | Group C |
|---|---|---|---|
| Feed type | *Elodea* weed | Dry lettuce, fish flakes, eggshell powder | Algae wafers, and trout pellets |
| Water type | Dam water (source water) | spring water | Dechlorinated water |
| Egg laying substratum | *C pyprus* twigs | White polystyrene strips | White polystyrene strips |
| Initial number of snails per group | 30 | 30 | 30 |
| Number of dead snails | 20 | 4 | 6 |
| Number of surviving snails | 10 | 26 | 24 |
| Percentage mortality | 67 | 13 | 20 |
| Egg masses collected | 10 | 69 | 59 |
| Number of hatched juveniles (F1 snails) | 48 | 500 | 450 |

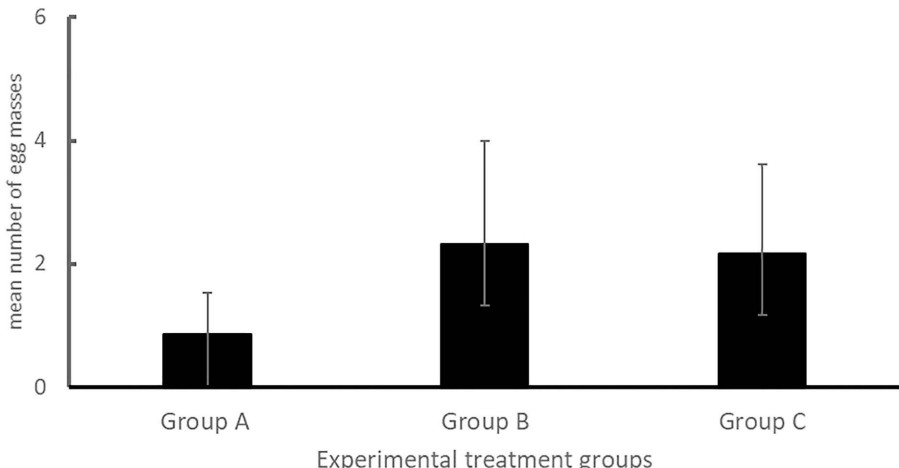

**Fig 1. Mean number of egg masses laid per group treatment.**

**Table 2. Regression outcomes of group treatment on egg masses.**

| Variable | Coefficient | Standard error | t | P | 95% CI |
|---|---|---|---|---|---|
| Groups | 0.6571054 | 0.3148534 | 2.09 | 0.047* | 0.0086526 −1.305558 |
| Constants | 0.465311 | 0.6801611 | 0.69 | 0.499 | −0.9342869-1.867349 |

*Statistically significant at p<0.05.

### 3.2. Survival

Fig 2 displays the Kaplan-Meier survival plot. Experimental group A (blue line) had the highest mortality rate compared with experimental groups B (red line) and C (green line). Experimental group B (red line) had the highest survival rate. Overall, statistical differences in the survival time were observed ($p < 0.001$).

The Cox proportional hazards model reveals that snails from experimental groups B and C had a significantly lower risk of mortality compared to experimental group A (Table 3). When compared to experimental group A, which was the control, there was an 88.3% reduction in the mortality of snails in experimental group B (Hazard Ratio: 0.117) (Table 3). No

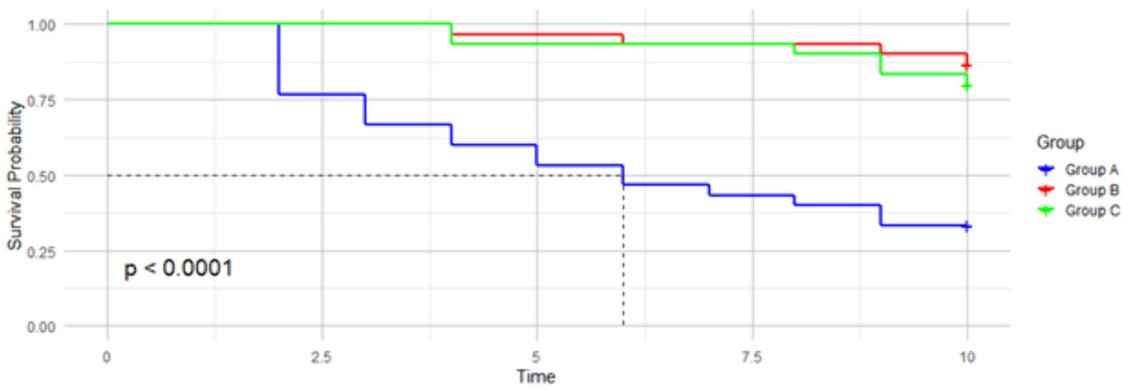

**Fig 2. Kaplan-Meier survival plot.**

**Table 3. Cox proportional hazards model results for snail mortality across treatment levels.**

| Treatments | Coefficient | Hazard Ratio | SE | z-value | p-value | 95%CI |
|---|---|---|---|---|---|---|
| Group B | −2.143 | 0.117 | 0.551 | −3.892 | <0.001* | (0.040-0.345) |
| Group C | −1.712 | 0.181 | 0.468 | −3.655 | <0.001* | (0.072-0.452) |

*Statistically significant at p<0.05.

difference in the survival rate was observed between snails maintained in experimental groups B and Group C ($p = 0.779$). The model's concordance statistic was 0. 749.

## 4. Discussion

Our findings highlight the critical role of laboratory conditions, particularly food type, water quality, and oviposition substrate in maintaining wild-caught $F_0$ generation *R. natalensis* and optimizing its reproductive output $F_1$ generation. Compared to the control (group A), our study showed that snails bred under the conditions of groups B and C produced a considerable number of egg masses and juvenile snails. Our improved protocol thus suggests that the experimental groups B or C approaches can be used, depending on the availability of resources for a particular laboratory. For laboratories from low-income settings that may seek to mass breed *R natalensis* for parasitological assessment, we believe that the protocol following the approach of the experimental C protocol would be the best because the resources needed, such as dechlorinated water, can be easily accessible compared to buying commercial spring water that was observed to produce the best results in this study. Despite numerical differences in survival, number of eggs laid, and juveniles hatching, the absence of a statistical difference between experimental groups B and C suggests that the outcome from the two approaches is comparable.

Many studies have described protocols for laboratory mass breeding of lymnaeid snails for improved understanding of the life history traits of *Fasciola* and potentially improving policy and enhancing livestock productivity [1,7,14,15,26,27]. However, these studies have also reported challenges in the breeding of *R. natalensis*, thus limiting our understanding of the ecology of these snails, especially in areas where climate change-induced emerging and re-emerging cases of fascioliasis occur. Hence, our study intended to improve the breeding protocols for *R. natalensis* under laboratory conditions. Our study considered previously published protocols for *R. natalensis* [7,14], *G. truncatula* [26], *and P. columella* [27] to come up with an improved and cost-effective version for mass snail production.

Our results showed that *Elodea*-fed snails in field water, which was designed to mimic the environmental conditions from which the snails were obtained, exhibited the lowest returns in terms of survival, egg masses laid, and number of juveniles laid. The high mortality rate and low fecundity observed in group A might be due to the natural water used for breeding that could have been contaminated by parasites that we did not screen and that may have interfered with the monitored parameters [28]. The potential presence of molds in the water and the vegetation that was used as substrate led to snail mortality. A study by Madson [7] suggests that mold affects adult snails, leading to a reduction in egg masses' hatchability and the survival of juvenile $F_1$ snails [7]. Snails in the wild are used to flowing water and a continuous replenishment of food, unlike in our study, where water was changed three times a week. The time lag in the changing of water may have also affected the feeding of snails due to poor water quality. The natural feed was inferior, particularly for protein content, to the artificial feed used for other groups. An earlier study suggested that *R. natalensis* prefers clean and clear water [28,29] and any slight change in the water quality affects snail survival. We also believe that the quality of the water may have been made worse by the *Elodea* sp. weed powder which was used as food and the *C. pyprus* twigs which were used as oviposition substrate. These factors might have been key in affecting snail survival and oviposition. These results highlight the importance of carefully selecting husbandry conditions in experimental protocols involving *R. natalensis*, particularly in studies on parasite life cycle maintenance.

## 5. Conclusions

This study underscores the trade-offs between biological efficacy and cost efficiency in the mass breeding of *R natalensis*. While snails in experimental group B demonstrated superior outcomes, laboratories in low-income settings may have a comparable number of snails if they chose to use the experimental group B approach for their work. This approach gives considerable number of eggs and juvenile snails at a minimum cost.

## Supporting information

**S1 File. Statistical analysis output for reproduction.**
(TXT)

**S2 File. Breeding data set.**
(XLSX)

## Acknowledgments

The administrative and technical assistance provided by Nokwanda Majola and Sambulo Gombela is gratefully acknowledged by the authors.

## Author contributions

**Conceptualization:** Agrippa Dube, Chester Kalinda, Tawanda Manyangadze, Moses John Chimbari.

**Data curation:** Agrippa Dube, Chester Kalinda, Tendai Makoni.

**Formal analysis:** Agrippa Dube, Chester Kalinda, Tawanda Manyangadze, Tendai Makoni, Moses John Chimbari.

**Funding acquisition:** Moses John Chimbari.

**Investigation:** Agrippa Dube, Tawanda Manyangadze, Moses John Chimbari.

**Methodology:** Agrippa Dube.

**Project administration:** Moses John Chimbari.

**Resources:** Moses John Chimbari.

**Supervision:** Chester Kalinda, Moses John Chimbari.

**Validation:** Moses John Chimbari.

**Visualization:** Chester Kalinda.

**Writing – original draft:** Agrippa Dube.

**Writing – review & editing:** Agrippa Dube, Chester Kalinda, Tawanda Manyangadze, Tendai Makoni, Moses John Chimbari.

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
