## [Decision Letter · Decision Letter 0]

27 Mar 2025

PONE-D-25-05169Standardized Protocol for Laboratory Rearing and Breeding of Radix natalensis Krauss, 1848: ResearchPLOS ONE

Dear Dr. Dube,

Thank you for submitting your manuscript to PLOS ONE. After careful consideration, we feel that it has merit but does not fully meet PLOS ONE’s publication criteria as it currently stands. Therefore, we invite you to submit a revised version of the manuscript that addresses the points raised during the review process.

Please find attached the reviewers’ reports on your manuscript. Both reviewers think that the study is a valuable contribution and that it needs restructuring for clarity and conciseness. The main criticism of reviewer #2 is about the absence of replicates for the experimental groups and she suggested that the experiments must be remade in order to assess the effects of the treatments more accurately.

We look forward to receiving your revised manuscript.

Kind regards,

Sthefane D`ávila

Academic Editor

PLOS ONE

2. We note you have included a table to which you do not refer in the text of your manuscript. Please ensure that you refer to Table 1 in your text; if accepted, production will need this reference to link the reader to the Table.

Reviewers' comments:

Reviewer's Responses to Questions

**Comments to the Author**

1. Is the manuscript technically sound, and do the data support the conclusions?

Reviewer #1: Yes

Reviewer #2: No

2. Has the statistical analysis been performed appropriately and rigorously? 

Reviewer #1: Yes

Reviewer #2: Yes

3. Have the authors made all data underlying the findings in their manuscript fully available?

Reviewer #1: Yes

Reviewer #2: Yes

4. Is the manuscript presented in an intelligible fashion and written in standard English?

Reviewer #1: Yes

Reviewer #2: Yes

5. Review Comments to the Author

Reviewer #1: This is a practice-oriented study, which can be of great interest of parasitologists studying trematod-snail interactions in Africa. The reviewed Ms is short and contains all the necessary information. I have only minor recommendations on how to improve the text.

1. As I can understand the design of the experiment, the group A was reared under conditions most similar to the natural one for R. natalensis. In this case, this group must be designated as the ‘control’ one, and all discussions of the results must be given as a comparison between the control groups and two other;

2. The reasons why the control group shows the worst performance (mortality, survival, fecundity, etc.) are not clear. The authors give reference to a number of papers, but these explanations are hardly applicable to laboratory conditions. For example, the authors cite several probable causes of low fecundity and survival, based on literary data: environmental changes, the presence of parasites, and inter and intraspecific competition between the snails and other aquatic microfauna. However, neither interspecific competition nor environmental changes are observed in a laboratory-reared culture consisting of individuals of the same species. I suppose that these causes should be deleted, and, instead, the authors have to try to explain, why in their experiment the control group was the worse.

3. An important recent paper on the laboratory rearing of lymnaeid snails has been overlooked by the authors: Dreyfuss, G., Vignoles, P., Rondelaud, D., Sánchez, J., Vázquez, A.A. (2023). Laboratory Cultures of Lymnaeidae for Parasitological Experiments. In: Vinarski, M.V., Vázquez, A.A. (eds) The Lymnaeidae. Zoological Monographs, vol 7. Springer, Cham. https://doi.org/10.1007/978-3-031-30292-3_14

Reviewer #2: The manuscript is of interest to researchers attempting to rear this snail species in the laboratory, a challenging task with lymnaeids, and is highly relevant due to the epidemiological significance of these species.

The most significant issue with this study is the lack of replication. Replicates are crucial in scientific experiments because they allow for the assessment of variability within treatments and ensure the reliability and generalizability of the results. In this study, the authors used 30 individuals per aquarium and assigned each aquarium a different treatment, with only 3 aquariums in total (and thus 3 treatments). This approach does not account for natural variation within each treatment group. Without replication, the results are less robust, and any observed effects may be due to random chance or uncontrolled environmental factors. Replicating treatments would provide more reliable data and allow for meaningful statistical comparisons, ultimately strengthening the conclusions of the study.

The manuscript should also be restructured to enhance clarity and conciseness, as it currently lacks these qualities. The authors should include information on studies involving other lymnaeids to provide a broader context and epidemiological data on this species to increase the value and interest of the work. Additionally, it is important to clarify the methods used by the authors to identify this snail species. It is not mentioned.

TITLE

I would delete “: Research”. I don’t understand the meaning.

Add the name of the family of this freshwater snail species.

ABSTRACT

The abstract needs to be restructured into a single paragraph without line breaks. Its current organization is unclear, and it should focus solely on presenting the main results concisely. Given the length of the manuscript, the abstract is too long and should be shortened.

INTRODUCTION

The introduction needs to be restructured following a well-known and straightforward approach for scientific writing: the inverted triangle technique. It should be organized as follows:

1. First paragraph – Provide general context on the rearing of lymnaeid snails and why laboratory breeding is essential for epidemiological studies.

2. Second paragraph – Highlight the lack of research on Radix natalensis and why studies on this species are particularly necessary. Discuss the impact of the disease it transmits, including its burden on livestock and human populations. Are there reported cases of fascioliasis in cattle or humans? If so, they should be mentioned to emphasize the relevance of studying this species.

3. Third paragraph – Clearly state what the authors did in this study, their objectives, and the methodology they used.

The introduction should be rewritten to improve readability and clarity, making it easier for the reader to follow the authors' reasoning.

Additionally, at one point, the authors refer to a study where experimental work with this species was conducted in the laboratory. However, it is unclear who conducted that study and how it relates to the present manuscript. If it is relevant, more context should be provided to clarify its significance.

The introduction should provide broader context on lymnaeid snails rather than focusing too much on Radix natalensis alone. While there is extensive research on other lymnaeid species, this particular species has not been studied as much, despite its significant epidemiological importance in this region of the world.

The authors should also revise and cite the following book chapter, which reviews laboratory breeding techniques for lymnaeid snails:

Dreyfuss, G., Vignoles, P., Rondelaud, D., Sánchez, J., Vázquez, A.A. (2023). Laboratory Cultures of Lymnaeidae for Parasitological Experiments. In: Vinarski, M.V., Vázquez, A.A. (eds) The Lymnaeidae. Zoological Monographs, vol 7. Springer, Cham. https://doi.org/10.1007/978-3-031-30292-3_14

Finally, in line 63, remove the period before the references.

M&M

This section also needs to be rewritten for clarity and conciseness. For example, the sentence: “Ninety adult field strains of Radix natalensis were collected from Kwafik'suthe Dam in Impendle and Mpofana municipalities in Mgungundlovu district, KwaZulu-Natal, South Africa.” is redundant, as this information is already provided in the previous paragraph. The number 'ninety' should be incorporated into the initial mention to avoid repetition.

Another example of redundancy is the sentence: 'Ninety snails in groups of thirty were assigned to three treatment Groups A, B, and C each comprising of 30 snails.' This phrase should be rewritten to avoid repetition and improve clarity.

The first subtitle of this section is Snail Collection and Identification; however, the authors do not address identification. This is a crucial aspect, as lymnaeid snails are often difficult to identify, and accurate species identification is essential to avoid erroneous conclusions. The authors should provide details on the identification methods used.

“The snails were fed three times per week, and water was changed three times a week. Snail mortalities and egg masses were recorded daily. The containers were washed three times per week to prevent accumulation of excretory products and excess food, and the water was changed afterward. Egg masses were recorded every day, and dead snail snails were removed.”. This passage contains redundancy and should be rewritten for clarity and conciseness. For example, the frequency of feeding, water changes, and cleaning is repeated unnecessarily. A more streamlined version would improve readability while maintaining all essential information.

RESULTS

Delete decimals in percentages. Use one decimal for values under 10% or over 90%. See http://adc.bmj.com/content/100/7/608

Figure 1 shows a deviation that I do not understand how it was calculated.

DISCUSSION

Given that there are no replicates for the treatments, I find it difficult to believe that the differences observed in the statistical estimates have biological relevance. Replicates are essential to ensure the reliability and generalizability of the results. Without them, it is challenging to draw meaningful conclusions about the true effects of the treatments.

To improve the experiment, the authors should include multiple replicates per treatment. For example, they could use at least three aquariums per treatment, each with 30 snails, to account for variability and ensure more reliable results. This way, the effects of the treatments can be more accurately assessed and statistically analyzed, leading to stronger conclusions.

6. PLOS authors have the option to publish the peer review history of their article (what does this mean?). If published, this will include your full peer review and any attached files.

Reviewer #1: No

Reviewer #2: No

---

## [Author Response · Author response to Decision Letter 1]

9 Apr 2025

I addressed all Editor and reviewer's comments in the the rebuttal letter and revised manuscript uploaded.

1. Editor comments on table 1 and the format were addressed. and the rebuttal letter addressed the reviewer's comments.

---

## [Decision Letter · Decision Letter 1]

15 Sep 2025

PONE-D-25-05169R1Standardized Protocol for Laboratory Rearing and Breeding of the lymnaeidae snail, Radix natalensis Krauss, 1848)PLOS ONE

Dear Dr. Dube,

Thank you for submitting your manuscript to PLOS ONE. After careful consideration, we feel that it has merit but does not fully meet PLOS ONE’s publication criteria as it currently stands. Therefore, we invite you to submit a revised version of the manuscript that addresses the points raised during the review process.

We look forward to receiving your revised manuscript.

Kind regards,

Clement Ameh Yaro, Ph.D

Academic Editor

PLOS ONE

Journal Requirements:

Reviewers' comments:

Reviewer's Responses to Questions

**Comments to the Author**

1. If the authors have adequately addressed your comments raised in a previous round of review and you feel that this manuscript is now acceptable for publication, you may indicate that here to bypass the “Comments to the Author” section, enter your conflict of interest statement in the “Confidential to Editor” section, and submit your "Accept" recommendation.

Reviewer #1: All comments have been addressed

Reviewer #2: All comments have been addressed

2. Is the manuscript technically sound, and do the data support the conclusions?

Reviewer #1: Yes

Reviewer #2: Yes

3. Has the statistical analysis been performed appropriately and rigorously? 

Reviewer #1: Yes

Reviewer #2: Yes

4. Have the authors made all data underlying the findings in their manuscript fully available?

Reviewer #1: Yes

Reviewer #2: Yes

5. Is the manuscript presented in an intelligible fashion and written in standard English?

Reviewer #1: Yes

Reviewer #2: No

6. Review Comments to the Author

Reviewer #1: (No Response)

Reviewer #2: The revised version of the manuscript has improved compared to the previous

submission. The authors have addressed several of the reviewers’ suggestions, and the

issue of replication has now been resolved. However, I believe further revisions are

needed to improve the grammar and overall writing style, in order to ensure that the

message is conveyed clearly and appropriately to the reader.

Below, I provide some examples of grammatical and stylistic corrections. Nevertheless,

I strongly recommend that the authors seek assistance from an English language editor

or a journal editor, as this would significantly enhance the clarity and readability of the

manuscript.

Abstract

• “Group A, containers were filled” → should be “Group A containers were filled”

• “0.85±0.22egg masses” → insert spaces: “0.85 ± 0.22 egg masses”

• “mass breading” → correct to “mass breeding”

• Sentence structure is dense; breaking long sentences would improve readability.

• Ensure consistency in terminology: use either “F1 offspring” or “F1 progenies”

throughout the manuscript.

Introduction

• The background section has been significantly improved and is now generally well-

structured, with a solid and relevant literature review.

• “Altitudinal differences suggest that understanding and evaluating breeding

feasibilities…” → Does this refers to differences in altitude between previous

studies and the current one?

• “Led to having inaccurate and not reproducible results” → replace by “led to

inaccurate and non-reproducible results”

• “Religions” → should be “regions”

• “Acclimazation” → replace by “acclimatization”

• “Conducting laboratory mechanistic experiments is in raising a snail colony…” →

rephrase for clarity: “A key requirement for conducting mechanistic experiments is

raising a clean, laboratory-bred snail colony…”

• The following sentence should be revised: “Breeding of snails is also important in

drug vaccine research [10]. Snails are used in trials to evaluate new anthelmintic

medications and biological control approaches, such as vaccines targeting different

parasite life cycle stages [11].” Vaccines against fascioliasis are currently being

developed for livestock (sheep and cattle), not for snails. Snails are not the target of

vaccination, and immune-based interventions in mollusks are not widely studied or

applied.

Materials and Methods

• Replace “R. natalensis snails for the study were identified…” by “Snails were

identified using morphological keys from Brown (1994) and Appleton (1996)...”

• “During the process of acclimatization to laboratory conditions and before being

used…” replace by “During acclimatization to laboratory conditions, 30 snails died

before allocation to treatment groups.”

• “10 transparent containers with 3 snails each…” replace by

“Each treatment group consisted of 10 transparent 2L plastic containers (26.65 cm ×

19.3 cm × 6.45 cm), with three snails per container, totaling 30 snails per group.”

• Treatments description: Improve clarity and reduce repetition: Group A (control):

fed Elodea weed powder; Cyperus papyrus twigs used as oviposition substrate;

field-collected water used; Group B: fed dry lettuce, fish flakes, and crushed

eggshells; spring water used; and Group C: fed algae wafers and trout pellets;

dechlorinated tap water used (left to stand for four days). All groups were fed three

times per week; water was changed at feeding to prevent waste buildup.”

• “Egg masses… were removed and transferred to new containers.” Replace by “Egg

masses laid on substrates or container walls were carefully removed and transferred

to new containers containing the same water type as the original group to ensure

consistency.”

• “Ambient and water temperatures were measured daily…” replace by “Ambient and

water temperatures were monitored daily and maintained at 26 ± 1 °C (air) and

22 ± 1 °C (water), respectively.”

Results

• “Snail egg-laying behaviour and hatching success were affected…” →replace by

something like “Egg-laying behavior and hatching success were influenced by food

type, water quality, and oviposition substrate.”

• “The snails in group A… laid the highest number of egg masses…” replace by

“Group A snails (fed Elodea weed powder, with papyrus twigs and field water)

produced the most egg masses (n = 91), followed by Group B (n = 57) and Group C

(n = 24).”

• “Overall, 197 egg masses comprising 1076 eggs…” replace by “A total of 197 egg

masses containing 1,076 eggs were laid across all groups: Group A accounted for

46.2% of eggs, Group B for 33.3%, and Group C for 20.3%.”

• “Hatching rates were highest in group A…” replace by “Hatching success was

highest in Group A (88.7%), moderate in Group B (75.6%), and markedly lower in

Group C (41.9%).”

Discussion

• “This study demonstrates the importance…” replace by something like “Our

findings highlight the critical role of laboratory conditions—particularly food type,

water quality, and oviposition substrate—in maintaining Radix natalensis and

optimizing its reproductive output.”

• “Snails fed Elodea…” replace by “Elodea-fed snails in field water exhibited

significantly higher reproductive output, suggesting this combination most closely

mimics their natural environment.”

• “The low egg production and hatching rate in group C…” replace by “The reduced

reproductive success observed in Group C likely reflects suboptimal nutritional and

water conditions, as neither the artificial feed nor dechlorinated tap water closely

replicate the snails’ natural habitat.”

• “This shows the need for care when designing…”. I suggest “These results

underscore the importance of carefully selecting husbandry conditions in

experimental protocols involving R. natalensis, particularly for studies on parasite

transmission or life cycle maintenance.”

7. PLOS authors have the option to publish the peer review history of their article (what does this mean?). If published, this will include your full peer review and any attached files.

Reviewer #1: No

Reviewer #2: No

---

## [Author Response · Author response to Decision Letter 2]

4 Oct 2025

Queries raised by Reviewer 2 to were all addressed

---

## [Decision Letter · Decision Letter 2]

15 Oct 2025

Standardized Protocol for Laboratory Rearing and Breeding of the Lymnaeidae snail, Radix natalensis (Krauss, 1848)

PONE-D-25-05169R2

Dear Dr. Dube,

We’re pleased to inform you that your manuscript has been judged scientifically suitable for publication and will be formally accepted for publication once it meets all outstanding technical requirements.

Kind regards,

Clement Ameh Yaro, Ph.D

Academic Editor

PLOS ONE

Additional Editor Comments (optional):

Reviewers' comments:

Reviewer's Responses to Questions

**Comments to the Author**

1. If the authors have adequately addressed your comments raised in a previous round of review and you feel that this manuscript is now acceptable for publication, you may indicate that here to bypass the “Comments to the Author” section, enter your conflict of interest statement in the “Confidential to Editor” section, and submit your "Accept" recommendation.

Reviewer #2: All comments have been addressed

2. Is the manuscript technically sound, and do the data support the conclusions?

Reviewer #2: Yes

3. Has the statistical analysis been performed appropriately and rigorously? 

Reviewer #2: Yes

4. Have the authors made all data underlying the findings in their manuscript fully available?

Reviewer #2: Yes

5. Is the manuscript presented in an intelligible fashion and written in standard English?

Reviewer #2: Yes

6. Review Comments to the Author

Reviewer #2: (No Response)

7. PLOS authors have the option to publish the peer review history of their article (what does this mean?). If published, this will include your full peer review and any attached files.

Reviewer #2: No

---

## [Editor Report · Acceptance letter]

PONE-D-25-05169R2

PLOS ONE

Dear Dr. Dube,

I'm pleased to inform you that your manuscript has been deemed suitable for publication in PLOS ONE. Congratulations! Your manuscript is now being handed over to our production team.

Kind regards,

on behalf of

Dr. Clement Ameh Yaro

Academic Editor

PLOS ONE